evolution, ecology, cellular biology

terrestrialization, algae, oxygen, redox balance, NADH/NDH complex

**Authors for correspondence:**
S. B. Gould
e-mail: gould@hhu.de
W. F. Martin
e-mail: bill@hhu.de

# Adaptation to life on land at high O2 via transition from ferredoxin-to NADH-dependent redox balance

S. B. Gould[1], S. G. Garg[1], M. Handrich[1], S. Nelson-Sathi[2], N. Gruenheit[1], A. G. M. Tielens[3,4] and W. F. Martin[1]

[1]Institute for Molecular Evolution, Heinrich Heine University Düsseldorf, 40225 Düsseldorf, Germany
[2]Interdisciplinary Biology, Computational Biology Laboratory, Rajiv Gandhi Centre for Biotechnology, Thiruvananthapuram, India
[3]Department of Biochemistry and Cell Biology, Faculty of Veterinary Medicine, Utrecht University, Utrecht, The Netherlands
[4]Department of Medical Microbiology and Infectious Diseases, Erasmus University Medical Center, Rotterdam, The Netherlands

SBG, 0000-0002-2038-8474; SGG, 0000-0003-4160-5228; SN-S, 0000-0002-3346-0383; WFM, 0000-0003-1478-6449

Pyruvate : ferredoxin oxidoreductase (PFO) and iron only hydrogenase ([Fe]-HYD) are common enzymes among eukaryotic microbes that inhabit anaerobic niches. Their function is to maintain redox balance by donating electrons from food oxidation via ferredoxin (Fd) to protons, generating $H_2$ as a waste product. Operating in series, they constitute a soluble electron transport chain of one-electron transfers between FeS clusters. They fulfil the same function—redox balance—served by two electron-transfers in the NADH- and $O_2$-dependent respiratory chains of mitochondria. Although they possess $O_2$-sensitive FeS clusters, PFO, Fd and [Fe]-HYD are also present among numerous algae that produce $O_2$. The evolutionary persistence of these enzymes among eukaryotic aerobes is traditionally explained as adaptation to facultative anaerobic growth. Here, we show that algae express enzymes of anaerobic energy metabolism at ambient $O_2$ levels (21% v/v), *Chlamydomonas reinhardtii* expresses them with diurnal regulation. High $O_2$ environments arose on Earth only approximately 450 million years ago. Gene presence/absence and gene expression data indicate that during the transition to high $O_2$ environments and terrestrialization, diverse algal lineages retained enzymes of Fd-dependent one-electron-based redox balance, while the land plant and land animal lineages underwent irreversible specialization to redox balance involving the $O_2$-insensitive two-electron carrier NADH.

## 1. Introduction

Molecular oxygen ($O_2$) had a far-reaching impact on evolution. From about 2.7–2.5 billion years ago onwards, cyanobacteria started using $H_2O$ as the electron donor for a photosynthetic electron transport chain consisting of two photosystems connected in series [1,2], generating $O_2$ as a waste product of primary production. Before that, all life was anaerobic [3,4]. However, oxygenation of the planet did not occur quickly, as atmospheric oxygen concentrations remained low for almost 2 billion years [5,6] (figure 1).

Current findings have shown that the monophyletic origin of land plants, which occurred some 450 Ma [12,13], boosted $O_2$ accumulation to modern levels through massive carbon burial [9,10]. Eukaryotes arose roughly 1.8 billion years ago [14,15], from which it follows that the first 1.3 billion years of eukaryote evolution took place in low oxygen conditions [7] at atmospheric and marine $O_2$ levels comprising only a fraction—0.001–10%—of today's $O_2$

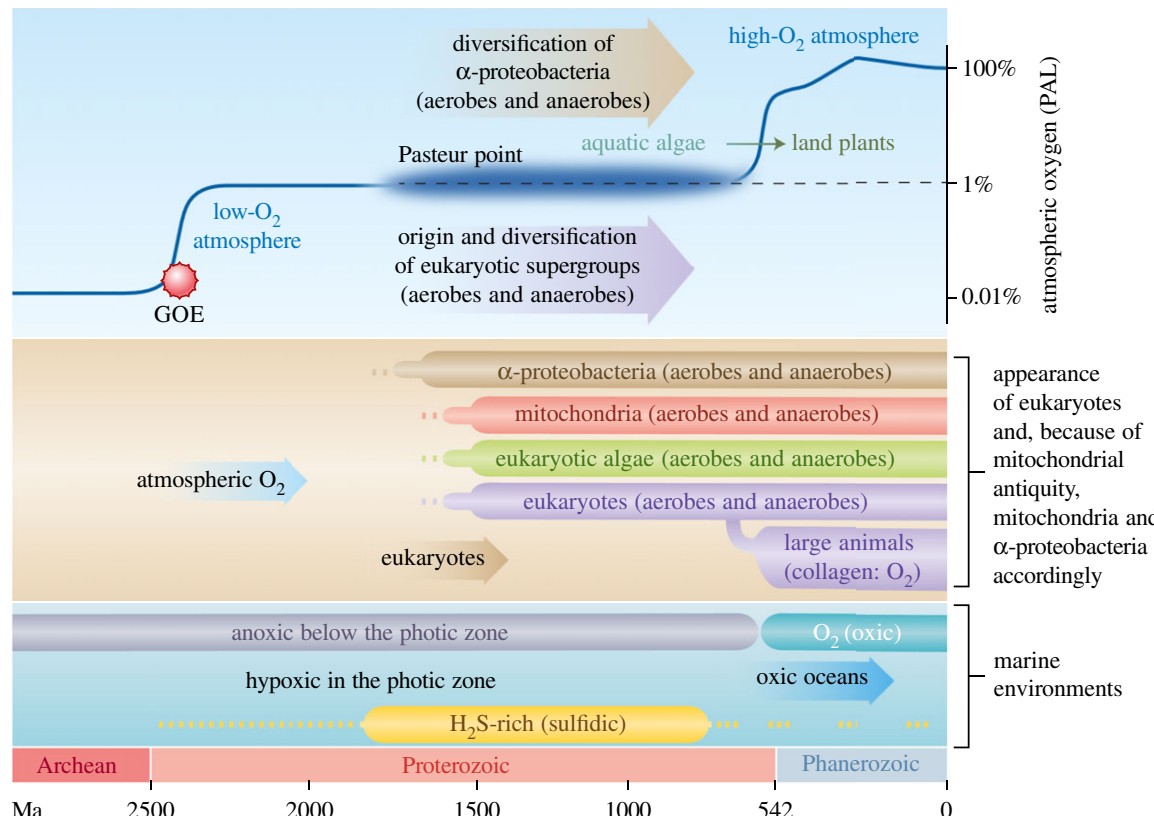

**Figure 1.** Overview of the changes in Earth's biochemistry and the rise and diversification of major groups with respect to oxygen concentration. After the great oxidation event (GOE) about 2.4 billion years ago, oxygen concentrations remained low, around the Pasteur point, as indicated by the cloudy line [2,5,7,8]. The rise of oxygen concentration to modern levels coincides with the conquering of land by streptophyte algae some 500 Ma and the beginning of massive carbon burial on land [6,9,10]. The Pasteur point is a classical term from microbiology that designates the $O_2$ partial pressure at which facultative anaerobes switch from aerobic to anaerobic metabolism (or vice versa); it corresponds to roughly 1% PAL or about 0.2% $O_2$ v/v [11]. Ma, million years ago. (Online version in colour.)

levels [5,6,9,10]. Because eukaryotes arose and diversified over a billion years before atmospheric $O_2$ reached the current value of 21% [v/v], it is hardly surprising that all major lineages (or supergroups) of eukaryotes possess enzymes of anaerobic energy metabolism (figure 2) [7]. In diverse eukaryotic lineages, these enzymes afford redox balance during ATP synthesis in mitochondria, anaerobic mitochondria [17], hydrogenosomes [18,19] and the cytosol [20] without requiring the presence of $O_2$ as the terminal acceptor [7,21].

The enzymatic backbone of redox balance in anaerobic energy metabolism in unicellular eukaryotes is pyruvate : ferredoxin oxidoreductase (PFO) and [Fe-Fe] hydrogenase ([Fe]-HYD), which were first described for eukaryotes in studies of carbon and energy metabolism in trichomonad hydrogenosomes [18]. The ecophysiological function of these enzymes, together with a larger set of proteins widely distributed across eukaryotes (figure 2), is generally interpreted as affording growth without oxygen. Hence, they are typically designated as enzymes of anaerobic metabolism. Like the pyruvate dehydrogenase complex of human or yeast mitochondria, PFO performs oxidative decarboxylation of pyruvate, generating acetyl-CoA and transferring electrons to the 4Fe4S cluster of the one-electron carrier ferredoxin (Fd). To maintain redox balance from growth substrate oxidation, reduced Fd ($Fd_{red}$) is reoxidized by [Fe]-HYD, which donates the electrons to protons, generating $H_2$ gas that leaves the cell as a waste product. $Fd_{red}$ generated by PFO is a low potential one-electron carrier ($Fd_{ox}/Fd_{red}$  $E_0 = -420$ mV) that can readily transfer a single electron to $O_2$ generating the superoxide radical, $O_2^{-}$·[22,23] and reactive oxygen species (ROS).

ROS are potent cytotoxins, a reason why organisms that employ the soluble PFO-Fd-[Fe]-HYD electron transport chain avoid high $O_2$ environments. In addition, PFO and [Fe]-HYD are irreversibly inactivated by $O_2$. Accordingly, eukaryotes that employ PFO and [Fe]-HYD in energy metabolism typically inhabit low oxygen environments, with their possession of these enzymes being interpreted as niche specialization [20,24,25].

However, PFO, [Fe]-HYD and a larger suite of enzymes associated with anaerobic energy metabolism are also present in algae [7,26–28], phototrophic eukaryotes with plastids that generate $O_2$. Their presence in algae is known to enable facultative anaerobic growth in low oxygen environments [7,28], and their expression is observed to be upregulated in response to anoxia in algae [29,30], in the same way that fermentation enzymes are hypoxia-induced in higher plants [31]. However, the expression in $O_2$-producing algae of enzymes associated with anaerobic redox balance has not been studied under normoxic conditions. Here, we investigated gene expression data from eukaryotic algae grown at ambient $O_2$ levels (21% v/v) to better understand the physiology, function and evolutionary persistence of Fd-dependent enzymes for one-electron-based redox balance in algae.

## 2. Distribution of enzymes for anaerobic metabolism in eukaryotes

The distribution of 47 genes for enzymes involved in anaerobic energy metabolism [7] in 56 eukaryotes spanning the

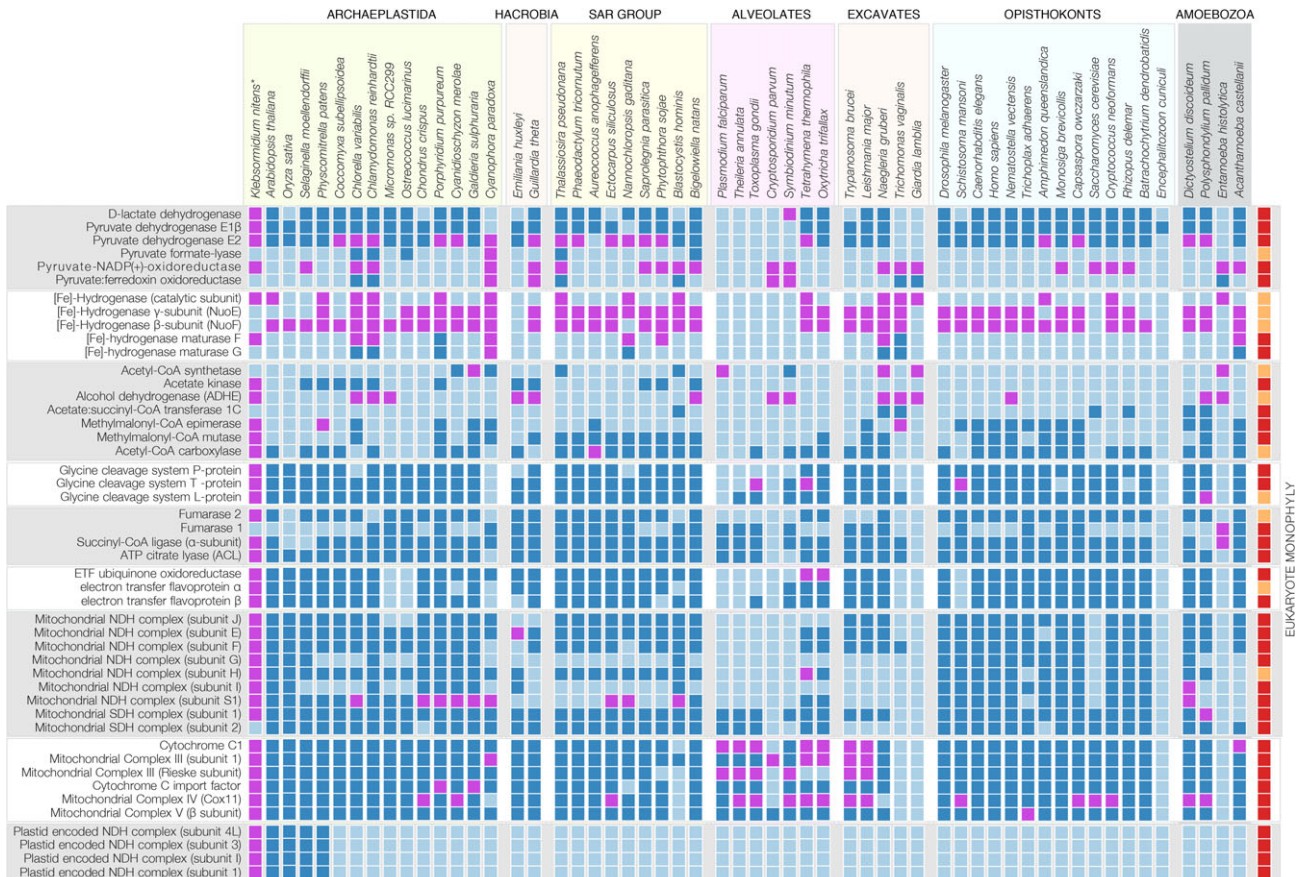

**Figure 2.** The presence–absence pattern of enzymes associated with anaerobic metabolism across the eukaryotic tree of life. The presence of each enzyme in eukaryotes scored as a dark blue square. An additional BLAST-based search (at least 30% identity and $e$-value of less than $10^{-7}$) identifies additional homologues (shown in magenta) that are not represented in the eukaryote–prokaryote clusters (EPCs) from Ku *et al*. [16] that is based, for example, on 40% global sequence identity for eukaryote proteins including BLAST hits for *K. nitens*, which was not included the original analysis [16]. Enzymes of anaerobic metabolism are present among all eukaryotic supergroups recognized, including all groups of algae, that is those carrying plastids of primary (e.g. *C. reinhardtii*, *Cyanaphora paradoxa*, *V. carteri*) and secondary origin (e.g. *B. natans*). For the enzymes that are identified as EPCs, phylogenetic trees (see the electronic supplementary material) indicate that 36 out of 43 (80%) of the genes show a single origin that traces to the eukaryotic common ancestor. Eukaryote monophyly as observed in phylogenetic trees constructed from protein sequences present in each cluster is shown with a dark red square (far right column), while orange indicates trees where the eukaryotes are non-monophyletic. For an extended presence–absence pattern including prokaryotes, see the electronic supplementary material, figure S1.

diversity of known lineages is summarized in figure 2. The enzymes are widely distributed across diverse eukaryotic lineages, although missing in some, consistent with a standard process of ecological specialization to aerobic and anaerobic habitats entailing the process of differential loss [16]. Some enzymes of one-electron-based redox balance have undergone lineage-specific functional specialization and have altered functional constraints in the protein. For example, [Fe]-HYD has lost its $H_2$-producing enzymatic activity in several eukaryotic lineages and has assumed different functions. The [Fe]-HYD homologues IOP1/NAR1 repress the hypoxia-inducible factor-1α subunit (HIF1-α) in humans [32] and, furthermore, possess conserved functions in cytosolic FeS cluster assembly in the human and yeast [33,34]. Prokaryotes employ $O_2$-labile FeS clusters for $O_2$-sensing and signalling [35]. In land plants, the [Fe]-HYD homologue has relinquished enzymatic activity to become the oxygen sensor *GOLLUM* [36].

Prokaryotic [Fe]-HYD enzymes can be trimeric [37], with 24 and 51 kDa subunits associated with the catalytic 64 kDa subunit, which contains the $H_2$-producing site, the H cluster. The 24 and 51 kDa subunits allow the enzyme to accept electrons simultaneously from both NADH and Fd via electron

confurcation [37], affording redox balance for both Fd and NADH pools. Some eukaryotic [Fe]-HYD enzymes, including the one from *Trichomonas* hydrogenosomes, also possess the 24 and 51 kDa subunits [38], which are related to mitochondrial complex I subunits. They are thought to allow the eukaryotes in question to perform electron confurcation, facilitating redox balance via NADH-dependent $H_2$ production [7], which would be thermodynamically unfavourable in the absence of $Fd_{red}$ [37,39].

Intermediate states in the evolutionary transition from Fd-dependent, one-electron-based redox balance to NADH-dependent redox balance are observed. In various eukaryotic lineages, PFO has become fused to an FAD–FMN–NAD binding domain that converts the ancestrally Fd-dependent enzyme (one-electron transport) into an $NAD(P)^+$-dependent enzyme that transfers hydride (two-electron transport) to generate NADPH. This fusion, called PNO for pyruvate:$NADP^+$ oxidoreductase [40], is now known to be widespread among eukaryotes (figure 2) [7,25], and represents an evolutionary intermediate in the transition from Fd-dependent to NADH-dependent redox balance, in that electrons from the FeS clusters of the PFO domain are channelled directly to NAD(P)H, bypassing the generation of soluble $Fd_{red}$.

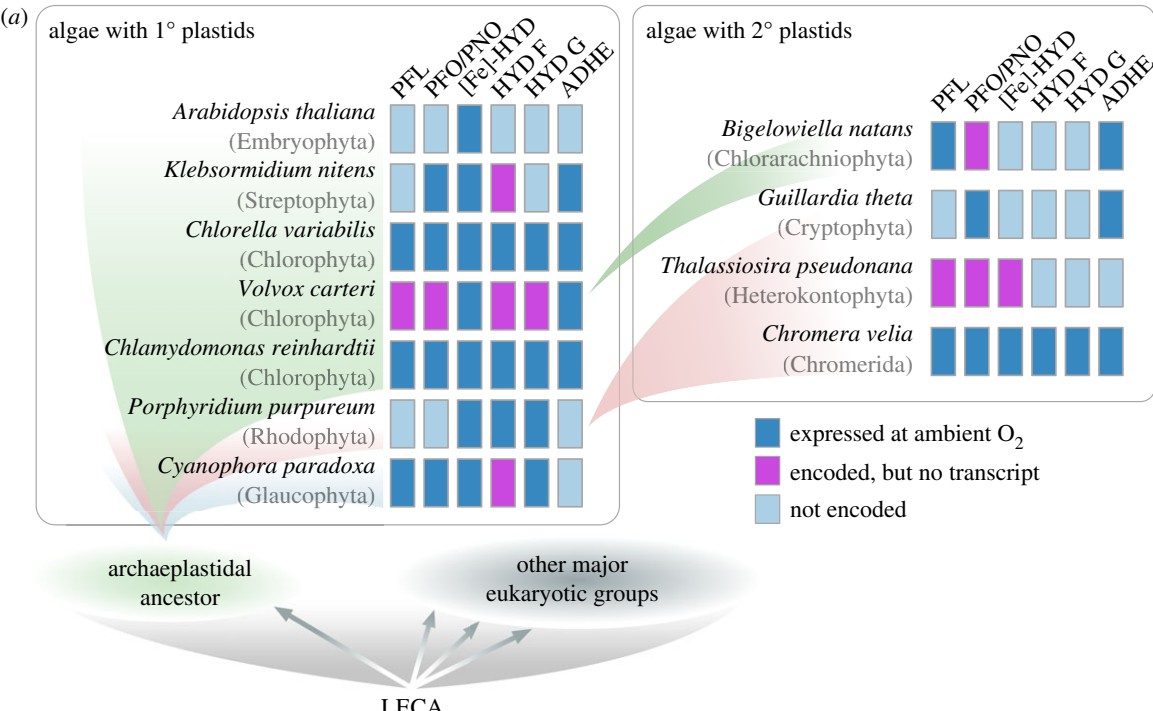

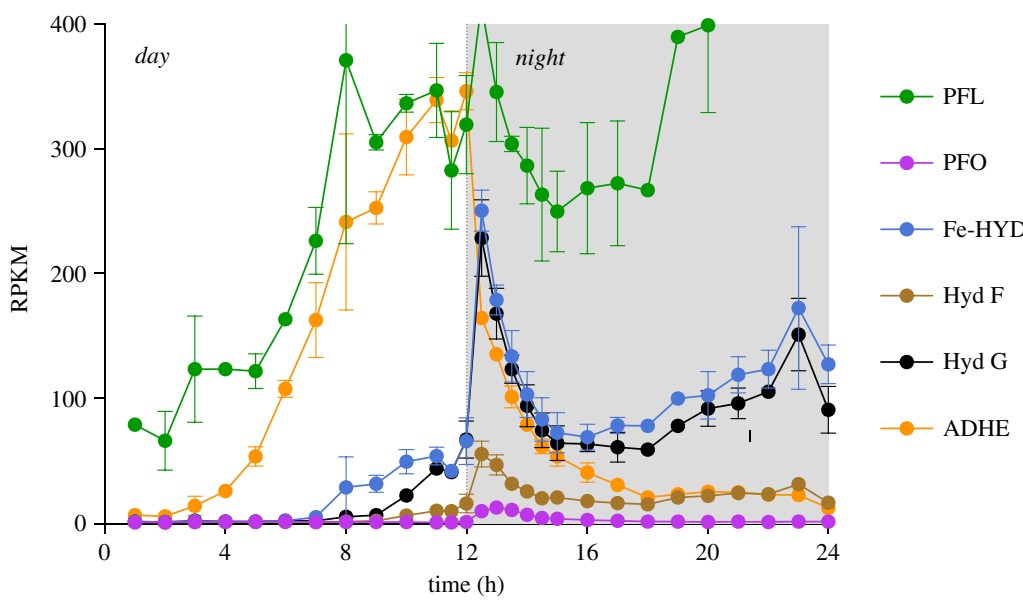

**Figure 3.** Expression of enzymes of anaerobic metabolism under aerobic conditions in algae and in a diurnal manner in *C. reinhardtii*. (*a*) Representative algae of all major groups, both with plastids of primary (Glaucophyta, Rhodophyta and Chlorophyta) and secondary origin (Cryptophyta and Chlorarachniophyta), are known to encode enzymes associated with anaerobic metabolism (figure 2). We found that those genes for enzymes of anaerobic redox balance are not only encoded in the algae, but are also expressed under ambient $O_2$ concentrations and independent of anaerobic growth conditions (except for *T. pseudonana*). (*b*) High-resolution RNA-Seq data of the chlorophyte alga *C. reinhardtii* show that the enzymes are mostly expressed in a diurnal manner under aerobic growth conditions. PFL is again seen to be expressed at high levels throughout (to the extent of a house-keeping gene), but with a peak early on during the dark phase that matches that of the other genes in question. LECA, last eukaryotic common ancestor; RPKM, reads per kilobase of transcript, per million mapped reads. (Online version in colour.)

## 3. Algae express enzymes for anaerobic metabolism at ambient $O_2$

The presence of the genes in representatives of the major algal groups (figure 2) raises the question of whether and when they are expressed. This is important, because genes for anaerobic energy metabolism have been retained in some eukaryotes with strictly $O_2$-dependent energy metabolism [41]. To determine whether enzymes of anaerobic redox balance are expressed independent of anaerobic culturing conditions, we generated transcriptome data for several

algal lineages with sequenced genomes: the red alga *Porphyridium purpureum*, the glaucophyte *Cyanophora paradoxa*, the chlorarachniophyte *Bigelowiella natans* with a plastid of secondary green origin and the cryptophyte *Guillardia theta* with a plastid of secondary red origin. All algae were grown under the same culturing conditions and at ambient $O_2$ levels of 21% [v/v]. In all algae studied, including algae with secondary plastids (figure 3*a*), we were able to detect the expression of at least a subset of the corresponding genes. It is well known that other algae such as *Vitrella brassicaformis* and *Chromera velia* encode a set of anaerobic

enzymes that is as complete as that of *C. reinhardtii* [25]. We therefore screened available transcriptome data for aerobically grown *Chr. velia* [42,43], *Volvox carteri* [44], *Chlorella variabilis* [45,46] and *Thalassiosira pseudonana* [47] and *Klebsormidium nitens* [48], and find that, for example, the chlorophyte *C. variabilis* and the chromerid *C. velia* (carrying a secondary plastid of red algal origin), express pyruvate formate lyase (PFL), PNO, hydrogenase maturases A/F/G (HydA/F/G) and bifunctional alcohol and aldehyde dehydrogenase (ADHE) in the same way as *C. reinhardtii* for which we generated RNA-Seq data (figure 3a).

The high-resolution RNA-Seq data available for *C. reinhardtii* [49] provided detailed insights into expression of enzymes for redox balance over the time course of 24 h. *Chlamydomonas* is among the algae that has preserved the most complete repertoire of $O_2$-sensitive enzymes involved in redox balance among eukaryotes studied so far (figure 2); it expresses them in the presence of 21% oxygen and in a diurnal fashion (figure 3b). PFL is found to be constantly expressed, but more so during the dark phase and in particular towards the end of the night (consistent with our RNA-Seq data). The same pattern is observed for its activating enzyme, although at much lower levels, similar to what is observed in prokaryotes [50]. Other genes in question, including both genes for the [Fe]-HYD catalytic subunit, HydA1 and HydA2, are upregulated with the onset of night (figure 3b). Importantly, this induction is observed independent of anaerobic culturing conditions, the standard method employed to induce [Fe]-HYD expression, typically in the context of biohydrogen applications [51–54]. The *Chlamydomonas* relatives *Chlorella* and *Volvox* display similar induction of enzymes involved in $H_2$ production and dark fermentation [55,56]; hence, anaerobiosis-independent expression is conserved and *Chlamydomonas* is the rule, not an exception.

The main finding from figure 3 is that the expression of the enzymes for anaerobic redox balance in eukaryotes does not correspond to any form of adaptation to anaerobic niches, as ambient $O_2$ does not change during the 24 h cycle. Instead, their expression in *C. reinhardtii* corresponds to the onset and end of illumination, where electron flux to and from the photosynthetic electron transport chain undergoes transient changes. In *Chlamydomonas*, PFO and [Fe]-HYD are localized in the plastid [54], not the mitochondrion or the cytosol, where they help to buffer electron flow into and out of the thylakoid membrane. This function does not preclude the existence of other functions under other conditions. For example, the same genes are expressed in *C. reinhardtii* during anaerobiosis [29,30]. Yet, for most of the algae surveyed in figure 3a, extended anaerobic growth phases are unknown, and the main habitat is the photic zone, where daily diurnal light conditions are encountered.

Some might view *C. reinhardtii* as an extreme case among algae, as it appears to mimic true anaerobic protists such as *Trichomonas* or soil-dwelling anaerobic bacteria when experiencing hypoxia. But *Chlamydomonas* can only endure anaerobic conditions for a limited amount of time, not thrive under them. To produce $H_2$ in a biofuels context, the typical procedure is to let *Chlamydomonas* cells assimilate $CO_2$ normally, then expose them to anoxic conditions while blocking photosystem II (PSII) to induce $H_2$-generating assimilate fermentation [29]. Low-level $H_2$ production by *Chlamydomonas* for up to two weeks can, however, be achieved under low-light conditions without PSII inhibition

[57]. This indicates that photosynthetic redox balance and one-electron-based redox balance conferred by the soluble PFO-Fd-[Fe]-HYD electron transport chain can operate independent of anaerobiosis. Finally, *C. reinhardtii* is not the only alga encoding such a complete set of anaerobic enzymes [25,28], but the only one that has been extensively studied in this respect.

## 4. Discussion

The retention and anaerobiosis-independent expression of Fd-dependent enzymes in algae, together with their localization to plastids in cases studied to date, indicates that the enzymes have been retained during algal evolution as the result of selection for redox balance in cells with one-electron transport. In terms of gene distribution (figure 2) and phylogeny (electronic supplementary material, figure S1), the enzymes of anaerobic energy metabolism in eukaryotes trace to the eukaryote common ancestor [17,26,28] (figure 2); hence, the archaeplastidan founder lineage that acquired the cyanobacterial ancestor of plastids already possessed them.

Eukaryotic enzymes involved in Fd-based redox balance have been the subject of many evolutionary investigations. There are two alternative hypotheses to account for their presence in eukaryotes. One has it that the Fd-dependent enzymes were present in the eukaryote common ancestor, which was a facultative anaerobe that was able to survive with or without $O_2$ as terminal acceptor, and were involved in its energy metabolism and redox balance [7,17,20,58]. The alternative lateral gene transfer (LGT) hypothesis has it that the ancestral eukaryote was a strict aerobe, unable to survive under anaerobic conditions, the presence of the Fd-dependent enzymes in eukaryotes resulting from multiple LGTs during eukaryote evolution to confer the ability to colonize anaerobic niches [25,59,60]. Directly at odds with the LGT theory is the observation that the archaeplastidal ancestor, whose PFL and PFL-activating enzyme are of monophyletic origin [59], did not adapt to an anaerobic niche, rather it acquired a cyanobacterial endosymbiont that became an $O_2$-producing plastid. The archaeplastidal lineage diversified into three main groups, representatives of which have retained the enzymes [25,28] (figure 2).

Though various formulations of the LGT hypothesis for enzymes of anaerobic redox energy metabolism in eukaryotes differ with respect to the number, nature and direction of LGTs [25], the underlying evolutionary rationale of the LGT hypothesis has remained constant: the lateral acquisition of Fd-dependent enzymes supposedly allowed eukaryotes to colonize oxygen-poor niches [61]. Notwithstanding the circumstance that the majority of eukaryote evolution occurred in oxygen-poor environments [5,7,9,20] (figure 1), the diurnal expression of Fd-dependent enzymes in algae at 21% [v/v] $O_2$ (figure 3) and independent of anaerobic growth conditions is incompatible with the view that the presence of these genes has anything to do with lateral acquisitions for adaptation to anaerobiosis. Rather, the data indicate that the genes for Fd-dependent redox balance were present in the eukaryote common ancestor, lost in some lineages during specialization to permanently oxic habitats (electronic supplementary material, figure S2) and retained in lineages that did not undergo the irreversible adaptation to complete $O_2$ dependence and NADH-dependent redox balance (figure 1).

Evolutionary responses to redox balance in eukaryotes can include recompartmentalization of pathways [62] to the cytosol, to plastids [63], to glycosomes [64] or to mitochondria [65]. Based upon their presence in the eukaryote common ancestor and their current localization within plastids in algae studied to date, the Fd-dependent enzymes PFO and [Fe]-HYD were recompartmentalized to the plastid during algal evolution. In the plastid, they assumed essential roles in light-dependent redox balance in an organelle that, upon contact with light, has no choice but to commence photosystem I (PSI)-dependent Fd reduction, rapidly depleting the available $Fd_{ox}$ pool. In land plants, $Fd_{red}$ is mainly reoxidized by ferredoxin : $NADP^+$ oxidoreductase (FNR), NADPH being reoxidized in turn by $NADP^+$-dependent glyceraldehyde 3-phosphate dehydrogenase (GAPDH) [66] in the Calvin cycle. In aquatic environments, $CO_2$ is more limiting than in air, for which reason algae have evolved diverse $CO_2$ concentrating mechanisms [67]. Algae thus require a means in addition to $CO_2$ fixation for redox balance at the onset of illumination, Fd-dependent enzymes of anaerobic energy metabolism fulfil that role. That functional aspect, redox balance in the plastid rather than anaerobiosis, accounts for diurnal expression and retention of enzymes for anaerobic redox balance among many independent algal lineages (figure 1). The expression of ferredoxin-dependent enzymes thus enables redox balance in the presence of $O_2$ in plastids and in the absence of $O_2$ as it occurs in *C. reinhardtii* ([29]) and many lineages of anaerobic protists that arose and diversified before the origin of plastids [7,20].

The transition to life on land approximately 450 Ma marked the advent of life in very high $O_2$ conditions [9,10]. Plants were the first major colonizers of land [68]. Massive carbon burial by land plants precipitated the high $O_2$ environment into which the first land animals followed (electronic supplementary material, figure S2). The colonization of land was, physiologically, an adaptation to high $O_2$ air. That adaptation to high $O_2$ witnessed the loss of Fd-dependent redox balance independently in both the land plant and land animal lineages (electronic supplementary material, figure S2) in response to the $O_2$ sensitivity of FeS clusters in PFO and [Fe]-HYD and in response to the ROS generating potential PFO of $Fd_{red}$. Once on land, both the plant and animal lineages were subsequently confronted again with hypoxic environments in adaptations to aquatic environments. The corresponding adaptations did not, however, involve gene acquisitions via LGT, merely novel expression regulation for NADH-dependent enzymes involved in redox balance during hypoxic response. In plants, these responses include mainly ethanol fermentations in waterlogged roots [31,69,70]. In animals, the evolutionary responses include various pathways regulated by the hypoxia-induced factor HIF [71,72], and secondary adaptations to the aquatic lifestyle among various vertebrates [73,74]. In addition, many marine and soil-dwelling invertebrates independently evolved their own specialized strategies for redox balance [7], from opine accumulation in mussels [75] to rhodoquinone dependent short chain fatty acid excretion in worms [76]. The land plant and land animal anaerobiosis adaptation pathways are, however, always NADH-dependent.

The retention of the chloroplast encoded NADH dehydrogenase complex (cpNDH) specifically in the land plant lineage (figure 1) is noteworthy. The functional cpNDH complex is localized close to complex I in thylakoids, both in the cyanobacterium *Synechocystis* [77] and in land plants, where it supports the cyclic flow of electrons essential for PSI to properly perform photosynthesis [78,79]. Among genes in plastid DNA, the cpNDH genes have undergone the highest number of independent losses [80]. Their retention in the plastid was probably a prerequisite for the transition to life on land [48,68], because they have been retained by the plastid in all land plant lineages, indicating a strong functional constraint for maintaining redox balance in the organelle [81]. Land plants have recruited a cytosolic NADH-dependent GAPDH [82] and a cytosolic malate dehydrogenase [83] to plastids for NADH-based redox balance. Even the origin of photorespiration, a process central to NAD(P)H-dependent redox balance, can be understood as an evolutionary response to high $O_2$ in the transition to life on land [84]. Land plant thylakoids cannot, however, relinquish Fd-dependent one-electron transport altogether, because the structure and function of PSI strictly require a steady flow of single electrons from the FeS clusters of PSI to generate soluble $Fd_{red}$, the stromal levels of which are monitored in some photosynthetic lineages by the flavodiiron (FLV) proteins [85]. Our findings indicate that in the plant and animal lineages, terrestrialization entailed an irreversible physiological transition away from one-electron-based Fd-dependent redox balance towards NAD(P)H-dependent redox balance involving two-electron transfers. The underlying evolutionary mechanisms were gene expression changes, enzyme recompartmentalization and gene loss in adaptation to high $O_2$ levels. Algae retained the Fd-dependent pathway for Fd-dependent, one-electron-based redox balance in plastids, not for anaerobic growth.

# 5. Material and methods

## (a) Identification of homologous proteins

As part of a larger study [16], sequences from 55 eukaryotes and 1981 prokaryotes (1847 bacteria and 134 archaea) were clustered into protein families in order to identify eukaryotic proteins with prokaryotic homologues. This approach resulted in 2585 disjunct clusters that contain at least two eukaryotes and no less than five prokaryotes. Within these 2585 eukaryote–prokaryote clusters (EPCs) using existing annotations, we identified 42 clusters containing proteins involved in anaerobic energy metabolism, which were relevant for the current analysis (electronic supplementary material, table S1). Phylogenetic trees and results from the tests on eukaryote monophyly were taken directly from [16] (shown in the electronic supplementary material, table S1). For proteins that did not have an EPC, the same dataset was used to perform a BLAST search and only hits with an identity of greater than 30% and an *e*-value of less than $10^{-10}$ were considered and provided in the electronic supplementary material, table S2. All the sequences that were used to identify the EPCs and perform the BLAST search are provided in the electronic supplementary material, file S1 along with the BLAST hits.

## (b) Cultivation of algae, RNA isolation and transcriptomics

All algae were grown in their respective media (see SAG Göttingen or ncma.bigelow.org) in aerated flasks under controlled conditions at 22°C and illuminated with 50 µE under a

12L : 12D cycle. RNA was isolated from cells growing in the exponential phase and at 6 h into the day and 6 h into the night. RNA was isolated using either Trizol™ reagent (Thermo Fisher, cat. no.: 15596018) or the Spectrum™ Plant Total RNA Kit (Sigma Aldrich, cat. no.: STRN50) according to the manufacturer's protocols. Then, samples were DNase treated (DNase I, RNase free, Thermo Fisher, cat. no.: EN0525) and RNA-Seq was performed at the Beijing Genome Institute (BGI, Hong Kong) using an Illumina HiSeq2000 resulting in 150 bp paired-end reads. For each sample, three individual runs were performed and pooled. Raw reads were subjected to several cleaning steps. First, adapter sequences as well as reads containing more than 5% of unknown nucleotides or more than 20% of nucleotides with quality scores less than 10 were removed. Further, reads were processed using TRIMMOMATIC (v. 0.35) [86] by removing the first 10 nucleotides as well as reads which showed a quality score below 15. Additionally, poly-A/T tails ≥ 5 nt were removed using PRINSEQ-LITE (v. 0.20.4) [87]. Finally, only reads with a minimum length of 25 nt were retained. Trimmed reads were assembled using TRINITY (v. 2.2.0) [88] and resulting contigs were filtered for a minimum length of 300 nt using an in-house perl script. Subsequently, open reading frames (ORFs) were identified using TRANSDECODER (v. 3.0.1) (https://github.com/TransDecoder/TransDecoder/wiki). These ORFs were used for a BLAST with an identity cut-off of 30% against the genome of the respective organisms to verify their presence in the genome and to remove possible contaminations. Transcriptomes are available via the Sequence Read Archive of NCBI (https://www.ncbi.nlm.nih.gov/sra) with the accession number PRJNA509798.

Data accessibility. All data have been made available in the form of electronic supplementary material or are publicly available.

Authors' contributions. S.B.G., S.G.G. and W.F.M. conceived the idea for the manuscript. S.G.G., M.H., and N.G. constructed the presence–absence table and compiled the expression data. All authors contributed to the writing and illustration of the manuscript.

Competing interests. We declare we have no competing interests.

Funding. We gratefully acknowledge the funding of this work by the DFG (to S.B.G.; 267205415–SFB 1208), the ERC (W.F.M.; 666053) and the VolkswagenStiftung to S.B.G. and W.F.M. (both Life).

Acknowledgements. We thank Verena Zimorski for discussions and help with formatting the manuscript.

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
