## [Reviewer comments · Proceedings of the Royal Society B: Biological Sciences]

Review History

RSPB-2019-1491.R0 (Original submission)

Review form: Reviewer 1

Recommendation

Accept with minor revision (please list in comments)

Scientific importance: Is the manuscript an original and important contribution to its field?

Excellent

General interest: Is the paper of sufficient general interest?

Excellent

Quality of the paper: Is the overall quality of the paper suitable?

Excellent

Is the length of the paper justified?

Yes

Should the paper be seen by a specialist statistical reviewer?

No

Do you have any concerns about statistical analyses in this paper? If so, please specify them explicitly in your report.

No

It is a condition of publication that authors make their supporting data, code and materials available - either as supplementary material or hosted in an external repository. Please rate, if applicable, the supporting data on the following criteria.

Is it accessible?

Yes

Is it clear?

Yes

Is it adequate?

Yes

Do you have any ethical concerns with this paper?

No

Comments to the Author

For consideration in any revision, I offer the following comments and queries.

Line 71. The Pasteur Point. Please define this, here or in the text. Or both. Not everyone knows this term.

Lines 87-89. "Fdred is a low potential one electron carrier (E_0' ca. -400 to -500 mV) that can readily transfer a single electron to O_2 generating the superoxide radical, $O_2^{\cdot-}$ ". A reference is required. Please consider Misra, H.P., Fridovich, I. (1971) J. Biol. Chem. 246, 6886-6890. and/or Allen JF (1975) Biochemical and Biophysical Research Communications 66: 36-43. Instead of "(E_0' ca. -400 to -500 mV)" one could write, instead, "($F_{dox}/F_{dred} E_0' = -420$ mV)"

Fig.2

What does the orange colour mean in the column "eukaryote monophyly"?

Klebsormidium nitens*. Why the asterisk...?

PFO is also known as pyruvate synthase and catalyses a reaction analogous with alpha-ketoglutarate synthase or 2-oxoglutarate synthase (could be AKGFO or 2OGFO). Does any eukaryote possess the latter enzyme?

Line 113. entails. Or "that entail" could be replaced by "and have".

Line 158. Thanks for explaining PNO. PFL (line 183) and other rarer abbreviations will be more easily understood if defined and unpacked on first appearance. Or defined in a glossary. At present the reader has to go searching, and then guess what is meant.

Fig. 3(b). "enzymes are mostly expressed in a diurnal manner under aerobic growth conditions" Won't the conditions change with the light-dark cycle? How are aerobic conditions maintained? "In aerated flasks" (line 383) - how? Was oxygen measured?

Lines 233-235. "Sustained hydrogen production by Chlamydomonas for an elongated period of

time is only feasible when PSII provides the main source of reducing power (Scoma et al. 2014). " I'm not sure H₂ is produced while H₂O is being oxidised. The experimental protocol for H₂ production is to allow PSII to run to fix CO₂ and then to transfer cultures to an anoxic phase, with PSII inhibited in some way (e.g. sulfur deprivation), then PSI alone works by receiving electrons from reduced carbon stores stored in the previous phase. Some authors regard induction of hydrogenase as a "normally a temporary emergency survival mechanism used in an oxygen-deprived environment". E.g. https://en.wikipedia.org/wiki/Anastasios_Melis and Gonzalez-Ballester et al. (2010) *Plant Cell* 22: 2058-2084. Fig. 3 shows this is unlikely to be true.

Fig. 4 and related text. Line 336 et seq. The idea that 2-electron redox balance began to be required in land when we reached current atmospheric oxygen levels. This is new, to my knowledge, and a major inference from this work that could be better underlined. Likewise the otherwise mysterious occurrence of NDH genes in plastids specifically of land plants. But why could the protein subunits not be imported as precursors?

Line 331. "The conquer of land" Odd. "Conquest" or "conquering" is more usual.

It is also a far-reaching conclusion to say that aerobic eukaryotes arose and diversified at very much lower oxygen levels than today – below the Pasteur point. This will surprise many, and should be given higher prominence in my opinion.

Figs. 2 and S1. "Mitochondrial" denotes subcellular location. Of gene or protein? The "Mitochondrial NDH complex subunits" in Archaeplastida - could these also include subunits targeted to the plastid? *Chlamydomonas* certainly has plastid "Respiratory Complex I" activity – "F₁F₀". It is also thought to be part of a cyt b₆f complex with PSI, and involved in "Chlororespiration". See e.g. Table I and Fig. 2 of Johnson & Alric (2013) *Eukaryotic Cell* 12, 776–793.

Review form: Reviewer 2

Recommendation

Accept with minor revision (please list in comments)

Scientific importance: Is the manuscript an original and important contribution to its field?

Good

General interest: Is the paper of sufficient general interest?

Good

Quality of the paper: Is the overall quality of the paper suitable?

Good

Is the length of the paper justified?

Yes

Should the paper be seen by a specialist statistical reviewer?

No

Do you have any concerns about statistical analyses in this paper? If so, please specify them explicitly in your report.

No

It is a condition of publication that authors make their supporting data, code and materials available - either as supplementary material or hosted in an external repository. Please rate, if applicable, the supporting data on the following criteria.

Is it accessible?

Yes

Is it clear?

Yes

Is it adequate?

No

Do you have any ethical concerns with this paper?

No

Comments to the Author

The manuscript presents very interesting and useful new data (transcriptomes for various algae) and re-analyses of published data demonstrating that aerobic algae express genes typically associated with anaerobic metabolism under oxic conditions. This is an important result because it speaks to the ongoing debate about the functions and evolutionary histories of genes of anaerobic metabolism in eukaryotes.

Specific comments:

- Figure 2: The hallmark genes (PFO and [Fe]-HYD, as discussed in the manuscript) are distributed very patchily, while the distribution of many other enzymes involved in anaerobic metabolism is much broader. Do these broadly-distributed enzymes have functions other than anaerobic metabolism?

- Gould et al. show very nicely that anaerobic genes are expressed at high levels in a coordinated way in *Chlamydomonas*. While time series are not available for the other algae, it seems important to quantify the expression of the anaerobic genes (e.g., against the rest of the transcriptome) in these other groups: beyond presence/absence in the transcriptome, are the key genes expressed at high levels, or is expression low?

Minor:

- The manuscript is written very well throughout but there are occasional awkward moments, e.g. line 44: "has had far...", or "had a far...", or "had...impacts"?
- line 68: "Overview of"?

Decision letter (RSPB-2019-1491.R0)

16-Jul-2019

Dear Mr Garg:

Your manuscript has now been peer reviewed and the reviews have been assessed by an

Associate Editor. The reviewers' comments (not including confidential comments to the Editor) and the comments from the Associate Editor are included at the end of this email for your reference. As you will see, the reviewers and the Associate Editor are positive, but also raise some issues with your manuscript and we would like to invite you to revise your manuscript to address them.

Research ethics:

Use of animals and field studies:

If you wish to submit your data to Dryad (<http://datadryad.org/>) and have not already done so you can submit your data via this link [http://datadryad.org/submit?journalID=RSPB&manu=\(Document not available\)](http://datadryad.org/submit?journalID=RSPB&manu=(Document+not+available)), which will take you to your unique entry in the Dryad repository.

Please submit a copy of your revised paper within three weeks. If we do not hear from you within this time your manuscript will be rejected. If you are unable to meet this deadline please let us know as soon as possible, as we may be able to grant a short extension.

Best wishes,

Professor Hans Heesterbeek
mailto: proceedingsb@royalsociety.org

Associate Editor

Board Member: 1

Comments to Author:

I have enjoyed reading this study by Garg and collaborators. The manuscript is mature, analyses sound, and the conclusions interesting. I agree with some of the minor suggestions made by the two other referees. I'd also add that the acronym PAP (presence/absence pattern) in Figure 2 legend needs either declared or just replaced (it is never used again in the main text).

I'd like to congratulate the authors for their work.

Reviewer(s)' Comments to Author:

Referee: 1

Comments to the Author(s)

For consideration in any revision, I offer the following comments and queries.

Line 71. The Pasteur Point. Please define this, here or in the text. Or both. Not everyone knows this term.

Lines 87-89. "Fdred is a low potential one electron carrier (E_0' ca. -400 to -500 mV) that can readily transfer a single electron to O_2 generating the superoxide radical, $O_2^{\cdot-}$ ". A reference is required. Please consider Misra, H.P., Fridovich, I. (1971) *J. Biol. Chem.* 246, 6886-6890. and/or Allen JF (1975) *Biochemical and Biophysical Research Communications* 66: 36-43. Instead of " E_0' ca. -400 to -500 mV)" one could write, instead, "($F_{dox}/F_{dred} E_0' = -420$ mV)"

Fig.2

What does the orange colour mean in the column "eukaryote monophyly"?

*Klebsormidium nitens**. Why the asterisk...?

PFO is also known as pyruvate synthase and catalyses a reaction analogous with alpha-ketoglutarate synthase or 2-oxoglutarate synthase (could be AKGFO or 2OGFO). Does any eukaryote possess the latter enzyme?

Line 113. entails. Or "that entail" could be replaced by "and have".

Line 158. Thanks for explaining PNO. PFL (line 183) and other rarer abbreviations will be more easily understood if defined and unpacked on first appearance. Or defined in a glossary. At present the reader has to go searching, and then guess what is meant.

Fig. 3(b). "enzymes are mostly expressed in a diurnal manner under aerobic growth conditions" Won't the conditions change with the light-dark cycle? How are aerobic conditions maintained? "In aerated flasks" (line 383) - how? Was oxygen measured?

Lines 233-235. "Sustained hydrogen production by *Chlamydomonas* for an elongated period of time is only feasible when PSII provides the main source of reducing power (Scoma et al. 2014)." I'm not sure H_2 is produced while H_2O is being oxidised. The experimental protocol for H_2 production is to allow PSII to run to fix CO_2 and then to transfer cultures to an anoxic phase, with PSII inhibited in some way (e.g. sulfur deprivation), then PSI alone works by receiving electrons from reduced carbon stores stored in the previous phase. Some authors regard induction of hydrogenase as a "normally a temporary emergency survival mechanism used in an oxygen-deprived environment". E.g. https://en.wikipedia.org/wiki/Anastasios_Melis and Gonzalez-Ballester et al. (2010) *Plant Cell* 22: 2058-2084. Fig. 3 shows this is unlikely to be true.

Fig. 4 and related text. Line 336 et seq. The idea that 2-electron redox balance began to be required in land when we reached current atmospheric oxygen levels. This is new, to my knowledge, and a major inference from this work that could be better underlined. Likewise the otherwise mysterious occurrence of NDH genes in plastids specifically of land plants. But why could the protein subunits not be imported as precursors?

Line 331. "The conquer of land" Odd. "Conquest" or "conquering" is more usual.

It is also a far-reaching conclusion to say that aerobic eukaryotes arose and diversified at very

much lower oxygen levels than today – below the Pasteur point. This will surprise many, and should be given higher prominence in my opinion.

Figs. 2 and S1. "Mitochondrial" denotes subcellular location. Of gene or protein? The "Mitochondrial NDH complex subunits" in Archaeplastida - could these also include subunits targeted to the plastid? *Chlamydomonas* certainly has plastid "Respiratory Complex I" activity – "FQO". It is also thought to be part of a cyt b6f complex with PSI, and involved in "Chlororespiration". See e.g. Table I and Fig. 2 of Johnson & Alric (2013) *Eukaryotic Cell* 12, 776–793.

Referee: 2

Comments to the Author(s)

The manuscript presents very interesting and useful new data (transcriptomes for various algae) and re-analyses of published data demonstrating that aerobic algae express genes typically associated with anaerobic metabolism under oxic conditions. This is an important result because it speaks to the ongoing debate about the functions and evolutionary histories of genes of anaerobic metabolism in eukaryotes.

Specific comments:

- Figure 2: The hallmark genes (PFO and [Fe]-HYD, as discussed in the manuscript) are distributed very patchily, while the distribution of many other enzymes involved in anaerobic metabolism is much broader. Do these broadly-distributed enzymes have functions other than anaerobic metabolism?

- Gould et al. show very nicely that anaerobic genes are expressed at high levels in a coordinated way in *Chlamydomonas*. While time series are not available for the other algae, it seems important to quantify the expression of the anaerobic genes (e.g., against the rest of the transcriptome) in these other groups: beyond presence/absence in the transcriptome, are the key genes expressed at high levels, or is expression low?

Minor:

- The manuscript is written very well throughout but there are occasional awkward moments, e.g. line 44: "has had far...", or "had a far...", or "had...impacts"?
- line 68: "Overview of"?

Author's Response to Decision Letter for (RSPB-2019-1491.R0)

See Appendix A.

Decision letter (RSPB-2019-1491.R1)

30-Jul-2019

Dear Mr Garg

I am pleased to inform you that your manuscript entitled "Adaptation to life on land at high O₂ via transition from ferredoxin to NADH-dependent redox balance" has been accepted for publication in Proceedings B.

Open Access

Paper charges

Sincerely,

Professor Hans Heesterbeek
Editor, Proceedings B
<mailto:proceedingsb@royalsociety.org>

Appendix A

Associate Editor

Board Member: 1

Comments to Author:

I have enjoyed reading this study by Garg and collaborators. The manuscript is mature, analyses sound, and the conclusions interesting. I agree with some of the minor suggestions made by the two other referees. I'd also add that the acronym PAP (presence/absence pattern) in Figure 2 legend needs either declared or just replaced (it is never used again in the main text).

I'd like to congratulate the authors for their work.

AU: We thank the editor for these kind words. We also thank the editor and both referees for their swift handling of our manuscript. We have replaced the acronym PAP with "presence-absence pattern" in Line 109. We have addressed all the suggestions made by the referees and hope that they are satisfactory and sufficient.

Additionally, in order to adhere with the length restrictions of the Proceedings of the Royal Society B we have moved our Figure 4 into the Electronic Supplementary Material (now Figure S2).

Referee: 1

Comments to the Author(s)

For consideration in any revision, I offer the following comments and queries.

Line 71. The Pasteur Point. Please define this, here or in the text. Or both. Not everyone knows this term.

AU: This has been now clarified in the legend which has been updated. (Line 68). "The Pasteur point is a classical term from microbiology that designates the O₂ partial pressure at which facultative anaerobes switch from aerobic to anaerobic metabolism (or vice versa); it corresponds to roughly 1% PAL or about 0.2% O₂ v/v [88] (Fenchel and Finlay, 1995)."

Lines 87-89. "Fdred is a low potential one electron carrier (E⁰ ca. -400 to -500 mV) that can readily transfer a single electron to O₂ generating the superoxide radical, O₂⁻." A reference is required. Please consider Misra, H.P., Fridovich, I. (1971) J. Biol. Chem. 246, 6886-6890. and/or Allen JF (1975) Biochemical and Biophysical Research Communications 66: 36-43. Instead of "(E⁰ ca. -400 to -500 mV)" one could write, instead, "(F_{dox}/F_{dred} E⁰= -420 mV)"

AU: Thank you. We have now added the references suggested.

Fig.2

What does the orange colour mean in the column "eukaryote monophyly"?

AU: Yes, this was missing. This is now been clarified in Line 111 which now reads "*...dark red (far right column), while orange indicates trees where the eukaryotes are non-monophyletic...*"

Klebsormidium nitens*. Why the asterisk...?

AU: Klebsormidium nitens was not part of the analysis in Ku et al. 2015 from which the data supporting the blue squares and the orange/red squares in Figure 2 were obtained. However, given its importance in terrestrialization, we thought it prudent to include as part of the analysis. This is also been clarified in the legend in Line 104.

PFO is also known as pyruvate synthase and catalyses a reaction analogous with alpha-ketoglutarate synthase or 2-oxoglutarate synthase (could be AKGFO or 2OGFO). Does any eukaryote possess the latter enzyme?

AU: Based on our current knowledge and a quick survey of KEGG alpha-keto glutarate synthase or 2-oxoacid oxidoreductases (other than PFO) are not found among eukaryotes.

Line 113. entails. Or "that entail" could be replaced by "and have".

AU: This has been corrected. Much better. Thank you.

Line 158. Thanks for explaining PNO. PFL (line 183) and other rarer abbreviations will be more easily understood if defined and unpacked on first appearance. Or defined in a glossary. At present the reader has to go searching, and then guess what is meant.

AU: Thank you for pointing this out. The acronyms are now spelled out at their first appearance.

Fig. 3(b). "enzymes are mostly expressed in a diurnal manner under aerobic growth conditions" Won't the conditions change with the light-dark cycle? How are aerobic conditions maintained? "In aerated flasks" (line 383) - how? Was oxygen measured?

AU: The data in Fig. 3(B) was obtained from Zones et al. 2015. The passage on Line 383 refers to the data from Fig3 (A) in which case aeration was maintained by growing the cultures under shaking conditions (at 100 rpm). While oxygen concentrations were not reported in that study, the media was (not S-deprived which is used for anaerobic growth of Chlamy) and it also appears unlikely that with the light switching off, the O₂ concentration would rapidly (not to say directly) fall to or below the Pasteur point. There is every reason to assume that the O₂ concentrations were ambient.

Lines 233-235. "Sustained hydrogen production by Chlamydomonas for an elongated period of time is only feasible when PSII provides the main source of reducing power (Scoma et al. 2014). " I'm not sure H₂ is produced while H₂O is being oxidised. The experimental protocol for H₂ production is to allow PSII to run to fix CO₂ and then to transfer cultures to an anoxic phase, with PSII inhibited in some way (e.g. sulfur deprivation), then PSI alone works by receiving electrons from reduced carbon stores stored in the previous phase. Some authors regard induction of hydrogenase as a "normally a temporary emergency survival mechanism used in an oxygen-deprived environment". E.g. https://en.wikipedia.org/wiki/Anastasios_Melis and Gonzalez-Ballester et al. (2010) Plant Cell 22: 2058-2084. Fig. 3 shows this is unlikely to be true.

AU: Yes, this passage was very poorly worded, many thanks for flagging it. The revised passage now reads: "To produce H₂ in a biofuels context, the typical procedure is to let *Chlamydomonas* cells assimilate CO₂ normally, then expose them to anoxic conditions while blocking PSII to induce H₂-generating assimilate fermentation [26]. Low level H₂-production by *Chlamydomonas* for up to two weeks can, however, be achieved under low light conditions without PSII inhibition [55]. This indicates that photosynthetic redox balance and one electron based redox balance conferred by the soluble PFO-Fd-[Fe]-HYD electron transport chain can operate independent of anaerobiosis." If we read Scoma et al. carefully, we see that some O₂ is being produced during H₂ production, but mitochondrial respiration (and other forms of respiration in *Chlamydomonas*) is operating as well, keeping net O₂ quite low. The point that we wish to make does not hinge on quantitative considerations. At any rate, many thanks for flagging this.

Fig. 4 and related text. Line 336 et seq. The idea that 2-electron redox balance began to be required in land when we reached current atmospheric oxygen levels. This is new, to my knowledge, and a major inference from this work that could be better underlined. Likewise the otherwise mysterious occurrence of NDH genes in plastids specifically of land plants. But why could the protein subunits not be imported as precursors?

AU: We think this is best explained by the CORR hypothesis, hence we wrote in the original: "Their retention in the plastid was likely a prerequisite for the transition to life on land [46,66], because they have been retained by the plastid in all land plant lineages, indicating a strong functional constraint for maintaining redox balance in the organelle [79]. "

Line 331. "The conquer of land" Odd. "Conquest" or "conquering" is more usual.

AU: Thank you. Done.

It is also a far-reaching conclusion to say that aerobic eukaryotes arose and diversified at very much lower oxygen levels than today – below the Pasteur point. This will surprise many, and should be given higher prominence in my opinion.

AU: We have added citations to the statement, which is well-represented in the literature "Because eukaryotes arose and diversified over a billion years before atmospheric O₂ reached the current value of 21% [v/v], it is hardly surprising that all major lineages (or supergroups) of eukaryotes possess enzymes of anaerobic energy metabolism (Fig. 2) [13,18].

Figs. 2 and S1. "Mitochondrial" denotes subcellular location. Of gene or protein? The "Mitochondrial NDH complex subunits" in Archaeplastida - could these also include subunits targeted to the plastid? *Chlamydomonas* certainly has plastid "Respiratory Complex I" activity – "FQO". It is also thought to be part of a cyt b6f complex with PSI, and involved in "Chlororespiration". See e.g. Table I and Fig. 2 of Johnson & Alric (2013) *Eukaryotic Cell* 12, 776–793.

AU: This indicates the location of the protein. Getting into the (highly controversial) topic of chlororespiration in *Chlamy* is not our objective here. We hope that R1 will agree on this point if we do not get into that discussion.

Referee: 2

Comments to the Author(s)

The manuscript presents very interesting and useful new data (transcriptomes for various algae) and re-analyses of published data demonstrating that aerobic algae express genes typically associated with anaerobic metabolism under oxic conditions. This is an important result because it speaks to the ongoing debate about the functions and evolutionary histories of genes of anaerobic metabolism in eukaryotes.

Specific comments:

- Figure 2: The hallmark genes (PFO and [Fe]-HYD, as discussed in the manuscript) are distributed very patchily, while the distribution of many other enzymes involved in anaerobic metabolism is much broader. Do these broadly-distributed enzymes have functions other than anaerobic metabolism?

AU: This is an astute observation. Not all the enzymes shown in Fig.2 are strictly involved in anaerobic metabolism. However, we included those since they can be part of pathways involved in energy metabolism (and they are usually included when discussing the topic), which is the focus of our paper.

- Gould et al. show very nicely that anaerobic genes are expressed at high levels in a coordinated way in *Chlamydomonas*. While time series are not available for the other algae, it seems important to quantify the expression of the anaerobic genes (e.g., against the rest of the transcriptome) in these other groups: beyond presence/absence in the transcriptome, are the key genes expressed at high levels, or is expression low?

AU: The data was sourced from different paper that not always used identical (although aerobic) conditions. High-res RNA-Seq data also only exists for *Chlamydomonas*. For this reason we decided to show the data the way we did and for the other algae just highlight their general expression. This is, however, worthy of exploring and high-res RNA-Seq data for other species is something we might look into in the future.

Minor:

- The manuscript is written very well throughout but there are occasional awkward moments, e.g. line 44: "has had far...", or "had a far...", or "had...impacts"?
- line 68: "Overview of"?

AU: We apologize for the oversight. The manuscript has now been carefully checked for typos and has been corrected for grammar to the best of our knowledge.

We thank Referee 1 and 2 for their kind comments and careful attention to our manuscript. We hope we have addressed all concerns that were raised.